# Molecular Modeling of Epithiospecifier and Nitrile-Specifier Proteins of Broccoli and Their Interaction with Aglycones

**DOI:** 10.3390/molecules25040772

**Published:** 2020-02-11

**Authors:** Juan Román, Dorian González, Mario Inostroza-Ponta, Andrea Mahn

**Affiliations:** 1Departamento de Ingeniería Química, Universidad de Santiago de Chile, Avenida Libertador Bernardo O’Higgins 3363, Estación Central, Santiago 9170019, Chile; juan.romana@usach.cl (J.R.); dorian.gonzalez@usach.cl (D.G.); 2Departamento de Ingeniería Informática, Universidad de Santiago de Chile, Avenida Libertador Bernardo O’Higgins 3363, Estación Central, Santiago 9170019, Chile; mario.inostroza@usach.cl

**Keywords:** broccoli, epithiospecifier protein, nitrile-specifier protein, aglycones, molecular docking

## Abstract

Glucosinolates are secondary plant metabolites of *Brassicaceae*. They exert their effect after enzymatic hydrolysis to yield aglycones, which become nitriles and epithionitriles through the action of epithiospecifier (ESP) and nitrile-specifier proteins (NSP). The mechanism of action of broccoli ESP and NSP is poorly understood mainly because ESP and NSP structures have not been completely characterized and because aglycones are unstable, thus hindering experimental measurements. The aim of this work was to investigate the interaction of broccoli ESP and NSP with the aglycones derived from broccoli glucosinolates using molecular simulations. The three-dimensional structure of broccoli ESP was built based on its amino-acid sequence, and the NSP structure was constructed based on a consensus amino-acid sequence. The models obtained using Iterative Threading ASSEmbly Refinement (I-TASSER) were refined with the OPLS-AA/L all atom force field of GROMACS 5.0.7 and were validated by Veryfy3D and ERRAT. The structures were selected based on molecular dynamics simulations. Interactions between the proteins and aglycones were simulated with Autodock Vina at different pH. It was concluded that pH determines the stability of the complexes and that the aglycone derived from glucoraphanin has the highest affinity to both ESP and NSP. This agrees with the fact that glucoraphanin is the most abundant glucosinolate in broccoli florets.

## 1. Introduction

Glucosinolates (GLS) constitute a large group of non-volatile, nitrogen- and sulfur-containing secondary metabolites mainly found in *Brassicaceae* family [1,2]. Their chemical structure consists of a β-d-glucopyranose residue linked to a thiohydroximate-O-sulfonate group by sulfur bridge, and an alkyl, aralkyl, or indolyl side chain (R) [3,4].

GLS play important roles in plant defense against biotic stress [7]. GLSs are stored in vacuoles or in specialized S-cells [8,9], and are relatively stable in the intact plant. However upon cell disruption, either by the action of microorganisms, insects or herbivores, GLS are broken down resulting into different compounds, such as isothiocyanates (ITC), thiocyanates (TCN), nitriles (NIT), or epithionitriles (EPT) (Figure 1), depending on the chemical conditions [10,11,12]. The reaction proceeds through a two-step mechanism. First, GLSs are hydrolyzed by myrosinase to yield an unstable aglycone intermediate (thiohydroximate-O-sulfonate). Second, the aglycone turns into isothiocyanates, thiocyanates, epithionitriles or nitriles, depending on pH, cofactors and specifier proteins [13].

Specifier proteins act on aglycones stabilizing them into compounds with antimicrobial, insecticide, and nematicide activity, showing no hydrolytic activity on glucosinolates themselves [14,15]. The formation mechanism of these products has not been elucidated so far. Some authors point out that specifier proteins are enzymes that act on aglycones [13,16]. Nevertheless, this has not been validated through kinetic studies or aglycone-protein crystallization studies mainly because of aglycones instability [15]. Specifier proteins may have a role in balancing the defense system of plants against generalist versus specialist herbivores, through different hydrolysis products [10,17]. Specifier proteins are grouped according to the final reaction product. Thiocyanate-forming proteins (TFP) promote the formation thiocyanates, and occur in few plant species (*Coronopus didymus*, *Eruca sativa*, *Lepidium ruderale*, *Lepidium sativum*, and *Thlaspi arvense*) [15,18,19]. Thiocyanates formation occurs only from allyl-GLS, 4-methylthiobutyl-GLS and benzyl-GLS [15,20,21]. Nitrile-specifier proteins (NSP) promote simple nitrile formation [22,23,24]. Epithiospecifier proteins (ESP) are relatively labile and promote epithionitriles formation [14]. ESP has a molecular weight between 30 and 40 kDa and is stabilized by Fe^2+^. Epithionitriles formation requires the presence of a terminal double bond in the glucosinolate aglycone [15,22]. Some studies revealed that ESP also promotes the formation of simple nitriles from aglycones lacking a terminal double bond [6,13].

Epithiospecifier protein (ESP) was isolated from *Crambe abyssinica* for the first time in 1973 and named for promoting epithionitriles formation [14]. In later studies, ESP was purified from *Brassica napus* L. [16,25]. To date, only one study on broccoli ESP has been reported. In that study, the formation of sulforaphane or sulforaphane nitrile were investigated by measuring ESP activity isolated from broccoli cv. Packman. Also, broccoli ESP was cloned and expressed in *Escherichia coli* obtaining 43 kDa recombinant protein [6,13]. To date, only *A. Thaliana* ESP structure has been resolved by X-ray diffraction (PDB code 5GQ0) [26].

Nitrile-specifier proteins (NSP) has been poorly studied so far. One of the first studies was carried out with *Pieris rapae* (white cabbage, *Lepidoptera*: *Pieridae*) NSP [27]. Five nitrile-specifier proteins (AtNSP1 to AtNSP5) in *A. thaliana* were identified later [22,23,28]. Currently, there is only one structure for NSP1 from *Arabidopsis thaliana* resolved by X-ray diffraction [29]. No studies about molecular interaction between broccoli ESP and broccoli NSP with aglycones are available so far.

In broccoli (*Brassica oleracea* var. *italica*), the main glucosinolate is glucoraphanin (GFN), and its hydrolysis at neutral pH yields preferably sulforaphane (SFN) which is one of the most potent food-derived anticancer compounds and also exhibits antimicrobial activity [30]. These facts encouraged the study of the myrosinase-glucosinolate system in broccoli in recent years. Different strategies have been proposed to increase SFN content in processed broccoli minimizing the formation of competing compounds such as nitriles and epithionitriles, which offer no health beneficial effects to the consumer. The strategies include blanching [31], high-pressure treatment [32], and microwave processing [33]. However, the complete conversion of GFN into SFN has not been achieved so far. Knowing the mechanism of action of broccoli ESP and NSP on aglycones occurring in broccoli would help in designing better strategies to maximize the conversion of glucosinolates into health-promoting compounds. The high instability of aglycones hinders experimental measurements. Hence, in this work, we studied the interaction between broccoli ESP and NSP with aglycones occurring in broccoli through molecular simulation tools.

## 2. Results and Discussion

### 2.1. Identification of the Main Glucosinolates Found in Broccoli Inflorescences

In order to select the aglycones to perform the molecular simulations, the main glucosinolates present in broccoli inflorescences were identified. Figure 2A shows the chromatogram of glucosinolates found in broccoli inflorescences. Five peaks were identified, according to Table 1. Figure 2B presents the mass spectrum of peak 3, corresponding to glucoraphanin. Figure 2C shows the structures of the identified glucosinolates.

The MS/MS product ions obtained from the M− anions (Table 1; Figure 2B and Appendix A) revealed two groups of typical fragments: one associated with the common moiety of the glucosinolates and the other providing diagnostic ions for the identification of the variable side chain. Some fragmentations differed from those described in literature. The m/z 436.4 signal observed at peak 3 (t_R_ 6.1 min) fragmented mainly to m/z 371.9 (Figure 2B) due to the loss of the methyl sulfoxide moiety; this peak was identified as glucoraphanin. The intensity distribution of fragments from glucobrassicin and hydroxy-glucobrassicin differed from those described in the literature, probably because of the different instruments used and the different fragmentation energies used. In this study, 2-propenyl glucosinolate (2PROP-GLS) and 3-(methylsulfinyl)propyl glucosinolate (3MSOP-GLS) were also identified (Table 1). Leng et al. [40] identified seven glucosinolates (GLS) in broccoli sprouts, with glucoraphanin being the most abundant. Wang et al. [41] reported seven aliphatic GLS in broccoli seeds, with glucoraphanin being the most abundant. Other authors reported that the main GLS in broccoli is glucoraphanin, followed by hydroxyl-glucobrassicin [4,42,43]. Accordingly, the five GSL detected in broccoli florets agree with the results found in the literature.

### 2.2. Homology Modeling

The amino acid sequence of broccoli nitrile-specifier proteins (NSP) has not been reported to date. Therefore, a multiple alignment was performed using Clustal W 2.0.12 including sequences of homolog organisms, in order to build a consensus sequence. The homologous NSP sequences were obtained from *Arabidopsis thaliana* (Q9SDM9), *Brassica rapa* subsp. *pekinensis* (M4FIJ4), *Brassica napus* (A0A078G479), and *Brassica oleracea* var. *oleracea* (A0A0D3AD54) (Figure 3).

The consensus amino acid sequence of NSP consisted of 467 residues (Appendix A), which encode for a protein with an estimated molecular mass of 51.1 kDa and a pI of 5.8. The amino acid sequence of broccoli epithiospecifier (ESP) was retrieved from the UniProtKB database (UniProtKB code: Q4TU02), and it consists of 343 amino acid residues, with an estimated molecular mass of 37.7 kDa and pI of 5.3. Simple alignment between *A. thaliana* and broccoli sequences gave identities of 80% for NSP and 76% for ESP. Since identities of 80% or lower usually imply structural differences, in this work structural models for broccoli ESP and NSP were built.

### 2.3. Modeling and Refinement of NSP and ESP Structures

The three-dimensional structural models of nitrile-specifier proteins (NSP) and epithiospecifier (ESP) were built using the Iterative Threading ASSEmbly Refinement (I-TASSER) server 5.1 based on the consensus sequence derived for NSP and in the reported sequence of ESP (Q4TU02). Among the five models proposed by I-TASSER for each protein, those showing the highest C-score and TM-score higher than 0.5 were chosen. For the NSP model the C-score was 1.87, and TM- score was 0.98. For the ESP model, the C-score was 1.40, and the TM-score was 0.91. The NSP model (Figure 4A), showed an “all-β” topology with 40 β-sheets and 43 loops. The ESP model (Figure 4B), showed a β - barrel topology, with 29 β-sheets and 29 loops.

The stability and dynamic properties of ESP and NSP models were evaluated using molecular dynamics (MD) simulations. To identify the energetically stable conformations, MD simulation was observed up to 60 ns. The root-mean-square deviation (RMSD) values were examined during the 60 ns of simulation in order to check the structural stability of models. RMSD values of ESP and NSP backbone atoms reached the equilibrium state after 25 ns for both proteins. The average RMSD obtained for ESP was 0.30 (nm) after 25 ns, and remained stable throughout the course of the simulation (Figure 5A). The average RMSD of NSP was 0.25 (nm), and it remained stable after 50 ns simulation (Figure 5B).

The ESP and NSP models were validated by ERRAT and Verify3D tools. According to the analysis of error values for each residue made by ERRAT, the average overall quality factor for ESP and NSP models were 85.4% and 82.1%, respectively, which is considered reliable [44]. Additionally, according to Verify3D evaluation, 99% of the ESP residues and 94.4% of the NSP residues had a 3D-score higher than 0.2, which indicates residues are in stable positions. Accordingly, the structural models were accepted as valid [45]. The results of the validation (Appendix A) confirm that the model quality was satisfactory. The number of H-bonds in the structural models of ESP and NSP was determined during the molecular dynamic’s simulations, and are presented in Appendix A. In ESP, H-bonds number fluctuates between 200 and 230, while in NSP varied from 300 to 320, along with the 60 ns simulations. H-bonds number of ESP was lower than NSP because ESP is 124 residues shorter. The results suggest that H-bonds do not vary significantly along the 60 ns trajectory. NSP and ESP models were deposited in the Protein Model DataBase (PMDB) with the access code PM0082618 and PM0082617, respectively (https://bioinformatics.cineca.it/PMDB/). These models were used for molecular docking simulations.

### 2.4. Molecular Interaction of NSP and ESP with Aglycones

The molecular docking simulations were carried out at different pH (1.0, 3.0, 5.0, 7.0), based on the pH range reported for epithiospecifier (ESP) and nitrile-specifier protein (NSP) [16,21,22,23,24,28,30,31]. The five aglycones considered in this study (3MSOP, 2PROP, 4MSOB, 4OHI3M, and I3M, see Table 2) were chosen based on the precursor glucosinolates detected in broccoli inflorescences. Binding sites identification was made with the SiteMap tool of Schrödinger suite 1–2019 [46,47]. SiteMap scores above 1.000 are generally considered as possible binding sites [48]. ESP SiteMap score for the selected site was 1.092, and that for NSP was 1.106. Despite using structural models that are less reliable than crystallographic structures, conservation of the residues that belong to the substrate binding sites could be observed when comparing with ESP and NSP from *A. thaliana*.

Table 2 shows the binding affinity energies and affinity constants obtained for the protein-aglycone complexes at different pH. The stability of the docked complexes was analyzed based on affinity constants.

The interaction of NSP with aglycones derived from indolic glucosinolates resulted in an average binding affinity energy of −8.6 kcal/mol, showing no significant variations between ligands and between the different pH. The affinity constants were found in the range of 1700–3300 (M) for these ligands. The interaction of NSP with aliphatic aglycones resulted in affinity binding energies between −6.7 and −5.8 kcal/mol with no apparent effect of pH. Binding affinity energy was lower for the interaction with indolic aglycones, and the affinity constant was lower for these ligands in comparison with aliphatic aglycones, indicating that indolic aglycones complexes are more stable.

The binding affinity energy of the complexes between ESP and aglycones derived from indolic glucosinolates was in average lower than those of aliphatic aglycones. The interaction between ESP and 4OHI3M was affected by pH showing no clear tendency. The other complexes were apparently not affected by pH in terms of binding affinity energy. Affinity constants varied between aglycones and pH, and were higher for the NSP complexes, in comparison with ESP. In turn, the ligand that showed the highest affinity constants was 4OHI3M, for both NSP and ESP.

The NSP–4OHI3M complex showed the highest affinity constant (kA) at pH 7 (3300 (M)). For ESP, the highest affinity constant was 2660 (M), at pH 7 complexed with I3M ligand, and K_A_ of the ESP–I3M complex remained constant at pH 1, pH 3, and pH 5 (220 (M)). For 4MSOB, the highest K_A_ was obtained at pH 5 for NSP and at pH 3 for ESP, giving values of 82 and 21 (M), respectively. The binding affinity energy was lower for the interaction of NSP with 4MSOB than for ESP with 4MSOB, at all pH. The lowest affinity constant was observed for the ESP-2PROP complex, being two orders of magnitude below the interaction between NSP–4OHI3M at pH 7.

The results suggest that the 4MSOB complexes are stabilized at acid pH, since the highest affinity constants for ESP–4MSOB and NSP–4MSOB were obtained at pH 3 and pH 5, respectively. The same order of magnitude of affinity constants may indicate that both ESP and NSP promote the formation of sulforaphane nitrile from glucoraphanin, at acid pH values (3; 5) in the presence of Fe^2+^. These results agree with those reported by Kissen et al. [23] and Backenköhler et al. [49], who found that the pH range of action for NSP and ESP from *A. thaliana* was pH 2–5.

Molecular docking simulations were conducted with the ligand that showed the highest affinity constants (4MSOB), and pH was set at pH 5 for NSP and at pH 3 for ESP. Fe^2+^ was included as a cofactor in all simulations. Figure 6 shows the interacting residues in the binding site of (A) NSP at pH 5, and (B) ESP at pH 3, with 4MSOB. Table 3 shows the atomic distances in the docked structures. The residues of NSP and ESP that interact directly with 4MSOB are at less than 6 (Å). Non-bonded interacting residues are summarized in Table 3.

Residues at the binding site of NSP were Arg220, Glu322, Ile168, Ser432, and Leu321. Among them, Fe^2+^ interacts with Glu322 forming an ionic interaction (Figure 6A). In addition, Fe^2+^ interacts with the oxygen of the carboxyl group of Leu321. Trp329 and His343 residues surround the binding site, stabilizing the aglycone (Table 3). The NSP residues that interact with the sulfonate group are Ile168, Arg220, and Ser432. This agrees with other studies that inform arginine is responsible for the stabilization of the sulfonate group in NSP [23,29,49].

Residues at the binding site of ESP were Val30, Arg80, Ser83, Ser135, Ile28, and Met81. The oxygen of the carboxyl group of Ile 28 and Met81 interact with Fe^2+^. Simulations suggest that the Val30 amino group and the Ser83 side chain stabilize the sulfonate group (Figure 6B). However, a study of non-interacting residues located in the surrounding of the ESP binding site, showed that His322, Glu334, and Asp333 residues could interact with Fe^2+^, forming a possible binding triad (Table 3), as reported by several authors [17,49].

Brandt et al. [17] reported that in the case of ESP from *A. thaliana*, a Glu260 - Asp264 - His274 binding triad interacts with Fe^+2^ while Arg94 and Arg157 are suggested to stabilize the thiohydroxamate-O-sulfonate group. Another study proposed that Gly186 and Val244 are responsible for the stabilization of aglycone in *A. thaliana* ESP [26]. In *A. thaliana* NSP (AtNSP3), iron-binding and NSP activity depend on Glu386, Asp390, and His394 [49]. Studies carried out on *Lepidium sativum* TFP (LsTFP), *A. thaliana* ESP (AtESP), and *Thlaspi arvense* TFP (TaTFP) showed that Arg94, Lis46, and Lis211 are involved in sulfate moiety binding [17].

The results obtained in this study suggest that broccoli ESP and NSP form more stable complexes with the aglycone 4MSOB at acid pH (3–5) in the presence of Fe^+2^. These results are consistent with studies that indicate that the specifier proteins are iron-dependent and act at acid pH [6,17,24,50].

## 3. Materials and Methods

### 3.1. Plant Material

Broccoli (*Brassica oleracea* var. *italica*) heads (three days after harvesting) were purchased at the local market (Santiago, Chile) from a single supplier. Broccoli inflorescences were immediately processed for quantification of glucosinolates and their breakdown products.

### 3.2. Glucosinolates Identification

Identification of glucosinolates (GLS) was conducted by the method reported by Francisco et al. [38] through LC/UV-PAD/ESI-IT-MS^2^. Fifty milligrams of broccoli inflorescences were extracted in 1.5mL 70% methanol at 70 °C for 30 min with vortex mixing every 5 min. The samples were centrifuged (13,000× *g*, 15 min, 4 °C). The supernatants were collected, and methanol was completely removed using a rotary evaporator (Stuart RE-300, Chelmsford, United Kingdom) at 37 °C. The dry material obtained was dissolved in 1 mL ultrapure water and filtered through a 0.22 µm syringe PVDF filter (Millex Durapore^®^, Merck, Darmstadt, Germany). Separation of GSL was carried out in Agilent 1100 HPLC (Agilent Technologies Inc., Santa Clara, CA, USA) series equipped with a photodiode array detector (PAD) coupled to the ESI-IT Esquire 4000 electrospray-ion trap mass spectrometer (Bruker Daltonik GmbH, Karlsruhe, Germany). For the control of the HPLC system, the ChemStation for LC 3D program (Agilent Technologies Inc., Santa Clara, CA, USA) was used, and for the spectrometer control, the EsquireControl 5.2 program (Bruker Daltonik GmbH, Karlsruhe, Germany) was used. For the HPLC separation, a Zorbax Eclipse Plus C18 column of 250 × 4.6 mm, 5 μm, and 100 Å (Agilent Technologies Inc., Santa Clara, CA, USA) was used. The separation of 20 μL of the sample was carried out at room temperature. The mobile phase consisted of the mixture of (A) 0.1% trifluoroacetic acid (TFA) and (B) acetonitrile/TFA (99.9: 0.1). A linear gradient was used starting with 0% of B in 0–5 min, reaching 17% of B in 15–17 min, 25% of B in 22 min, 35% B in 30 min, 50% B in 35 min, 99% of B in 50 min and in 55–65 min 0% of B. The flow rate was 1 mL/min. Absorbance at 227 nm was recorded. The analysis of chromatograms and mass spectra was performed through the program Data Analysis 3.2 (Bruker Daltonik GmbH, Karlsruhe, Germany). All measurements were performed in triplicate. The structures of the detected glucosinolates were confirmed according to their retention times and m/z signal (species [M-H]^−^) (Appendix A), MS/MS fragmentation as well as from the characteristic product ions (Appendix A) [34,35,36,37,38,39,51,52,53,54,55,56,57]. The study of MS^2^[M−H]^−^ fragmentation of broccoli glucosinolates, shows specific product ions at m/z around 194, 242, 259, and 275, which correspond to the fragment ions from the aglycone side chain that have been reported by other authors [35,38,39].

### 3.3. Template Identification

The amino acid sequence of broccoli epithiospecifier protein (ESP) was obtained from UniProtKB protein sequence database (UniProtKB code: Q4TU02) [6]. To obtain the consensus amino acid sequence of broccoli nitrile specifier protein (NSP_BRO), the sequences Q9SDM9 (*A. thaliana*) [22], M4FIJ4 (*Brassica rapa subsp. pekinensis*) [58], A0A078G479 (*Brassica napus*) [59], and A0A0D3AD54 (*Brassica oleracea* var. *oleracea*) [60] were used as templates to perform multiple alignment.

### 3.4. ESP and NSP Modeling and Models Evaluation

Epithiospecifier protein ESP and nitrile-specifier protein (NSP) modeling was performed using the Iterative Threading ASSEmbly Refinement (I-TASSER) server 5.1 based on the ab initio/threading method [61,62]. I-TASSER uses a hierarchical protein structure modeling approach. A total of five three-dimensional (3D) models were generated for each protein (ESP and NSP); the model was selected based on the confidence score (C-Score), and validated by the evaluation tools ERRAT and Verify3D (available at http://services.mbi.ucla.edu).

### 3.5. Molecular Dynamics Simulation

Molecular dynamics simulations were carried out in GROMACS 5.0.7 [63,64,65]. The three-dimensional structure models of broccoli epithiospecifier protein (ESP) and nitrile-specifier protein (NSP) were selected based on highest C-score value (range of (−5.2)) and TM-score (higher than 0.5) given by Iterative Threading ASSEmbly Refinement (I-TASSER) [61,62,66] and were used for the MD simulation study with OPLS-AA/L all-atom force field (2001 amino acid dihedrals) [67]. An orthorhombic box with a predefined TIP3P water model was constructed around the 3D model. The box volume was minimized, and Fe^2+^ ions were added applying position restrains to the protein structure. The temperature and pressure of the system were kept constant at 298 K and 1.01325 bar after using the Berendsen weak-coupling method. A cut-off radius of 0.9 nm was used in the simulation of van der Waals and electrostatic interactions. Finally, a 60-ns production was carried out for ESP and NSP models using the particle mesh Ewald (PME) electrostatics method under NPT conditions. For analysis, final coordinates were saved every 20 ps, subjected to energy minimization. The refined model was used for molecular docking studies.

### 3.6. Molecular Docking

The binding affinities between epithiospecifier protein (ESP) and nitrile-specifier protein (NSP) and the aglycones derived from glucoraphanin (4MSOB-GLS), hydroxy-glucobrassicin (4OHI3M-GLS), glucobrassicin (I3M-GLS), glucoiberin (3MSOP-GLS), and sinigrin (2PROP-GLS) were obtained using AutoDock Vina 1.1.2 [68]. The aglycone structures were optimized by the LigPrep tool in Maestro of Schrödinger suite 1-2019 [69]. ESP and NSP models were pre-processed and prepared by Protein Preparation Wizard in Maestro of Schrödinger suite 1-2019. Protonation and tautomeric states of each protein and ligand were adjusted at different pH (1, 3, 5, and 7), using the Epik 2.1 ionization tool of the Schrödinger suite 1-2019 [70,71]. Finally, the receptor grid was located at the binding sites of ESP and NSP. The identification of protein binding sites was made with the SiteMap tool of Schrödinger suite 1-2019 [46,47]. The grid was a cubic box, centered at the centroid of the binding site residues, that in case of the ESP center was: X: 54.995; Y: 58.009; Z: 63.829 and its size was: X: 96, Y: 96, Z: 96. The center of NSP was: X: 64.252, Y: 65.860, Z: 77.247, and its size was: X: 80, Y: 80, Z: 80. For protein-ligand docking, the exhaustiveness was 15. The most stable docking orientation was identified from the binding affinity score and hydrogen bond interaction to the binding site based on visual inspection. Discovery Studio 3.5 visualizer (DS 3.5) (DS, http://www.accelrys.com; Accelrys, Inc. San Diego, CA) generated the 2D graphical visualization of ESP, NSP, and protein-ligand complexes.

## 4. Conclusions

Three-dimensional models of broccoli epithiospecifier protein (ESP) and nitrile-specifier protein (NSP) were constructed and deposited in Protein Models Database with the identifier PM0082617 and PM0082618, respectively. ESP model consisted of 29 β sheets and 29 loops, while the NSP model had 40 β sheets and 43 loops. Docking simulations showed that the NSP-4MSOB complex at pH 5 was more stable than ESP-4MSOB at pH3, in the presence of Fe^2+^. The NSP residues that interact with 4MSOB and Fe^2+^ were Arg220, Glu322, Ile168, Ser432, and Leu321. Simulations showed that Fe^2+^ interacted with Glu322 and with Leu321. Ile168, Arg220, and Ser432 interact with sulfonate group. Trp329 and His343, located around the binding site, would be responsible for aglycone stabilization. The residues that belong to ESP binding site were Val30, Arg80, Ser83, Ser135, Ile28, and Met81. Docking simulations suggested that Fe^2+^ interacts with Ile 28 and Met81. Val30 and Ser83 interact with the sulfonate group, and His322 - Glu334 - Asp333 binding triad was proposed. The complexes containing 4MSOB were stabilized at pH = 3 for ESP and pH = 5 for NSP. Finally, pH determines the stability of the complexes, and the aglycone derived from glucoraphanin (4MSOB) has the highest affinity to both ESP and NSP. This agrees with the fact that glucoraphanin is the most abundant glucosinolate in broccoli florets. Our results will probably help in understanding the mechanism of action of specifier proteins and aglycones from broccoli.

## Figures and Tables

**Figure 1 molecules-25-00772-f001:**
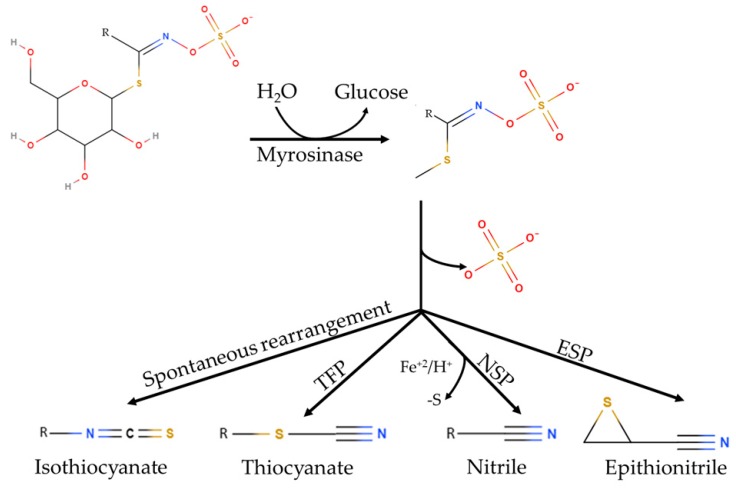
Glucosinolate hydrolysis. Glucosinolates are hydrolyzed by myrosinase yielding unstable aglycones. These aglycones can form isothiocyanates spontaneously, or other hydrolysis products, depending on the presence of specifier proteins (thiocyanate-forming proteins (TFP), epithiospecifier (ESP), nitrile-specifier proteins (NSP)) modified from [5,6].

**Figure 2 molecules-25-00772-f002:**
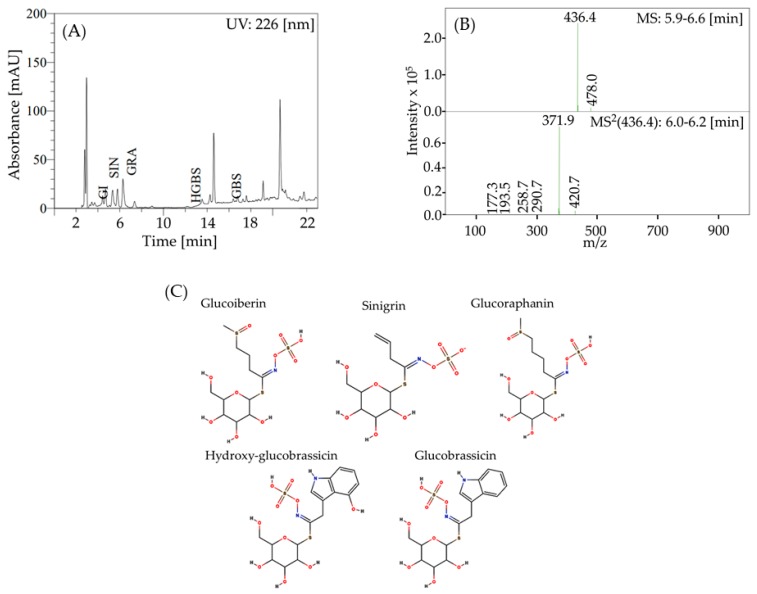
(**A**) High Performance Liquid Chromatography with Photo Diode Array (HPLC-PDA) chromatogram of glucosinolate profile of broccoli inflorescences. Detection at 226 nm. Peaks: (GI) glucoiberin, (SIN) sinigrin, (GRA) glucoraphanin, (HGBS) hydroxy-glucobrassicin, and (GBS) glucobrassicin. (**B**) Liquid Chromatography—Mass Spectrometry (LC-MS) spectra of glucoraphanin. (**C**) Structures of the major glucosinolates identified in broccoli. Appendix A shows the mass spectra of glucoiberin, sinigrin, hydroxy-glucobrassicin, and glucobrassicin.

**Figure 3 molecules-25-00772-f003:**
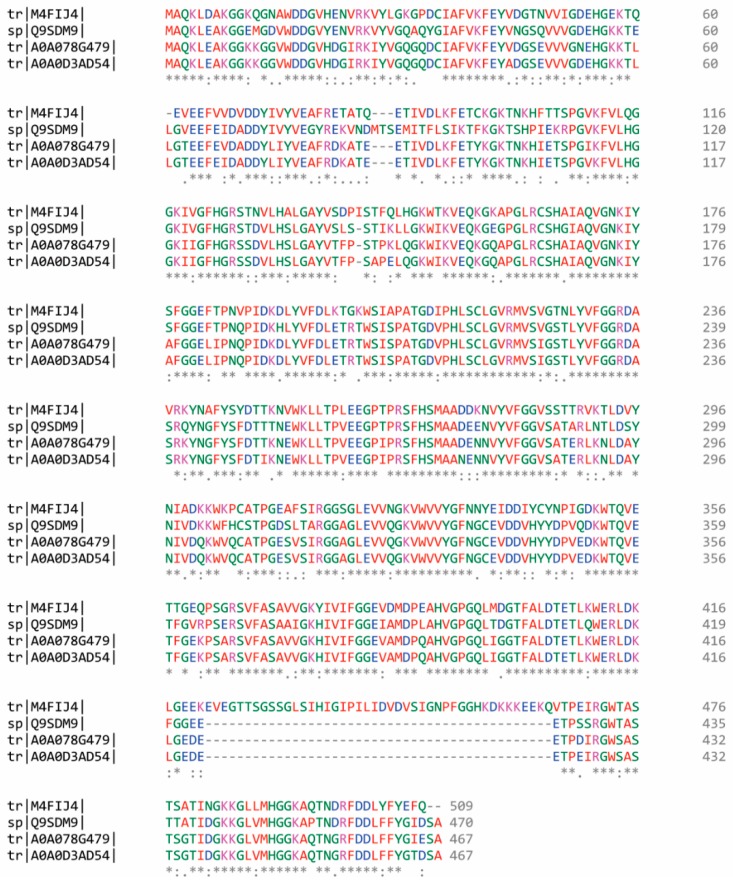
Multiple alignments of Nitrile-specifier Protein (NSP) from broccoli homolog organisms (Q9SDM9, M4FIJ4, A0A078G479, and A0A0D3AD54), generated by means of Clustal W. Gaps and conservation residues were denoted in dashes (-) and asterisks (*), respectively. In addition, colon (:) and period (.) indicate strong and weak properties conservation between groups, respectively. The color code represents the physicochemical property of the amino acid (red: small or hydrophobic, blue: acidic; magenta: basic; green: hydroxyl or sulfhydryl or amine; grey: unusual amino/imino acids).

**Figure 4 molecules-25-00772-f004:**
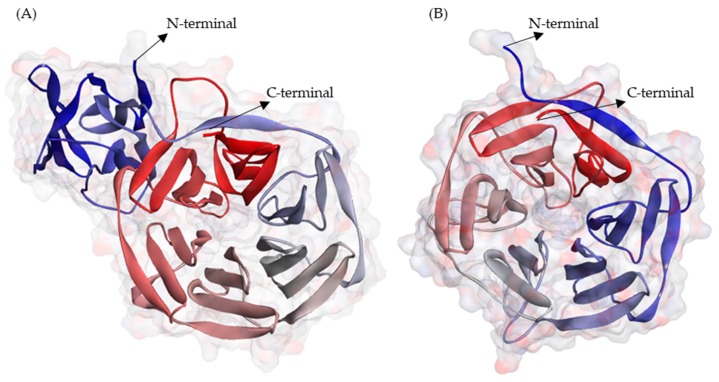
Three-dimensional structure models of (**A**) Nitrile-specifier Protein (NSP) and (**B**) Epithiospecifier Protein (ESP) from broccoli. The models were represented using DS 3.5 visualizer, and colors of N-to-C terminal are based on their secondary structure. The models are available at PMDB (https://bioinformatics.cineca.it/PMDB/) with the identifier PM0082618 for NSP and PM0082617 for ESP.

**Figure 5 molecules-25-00772-f005:**
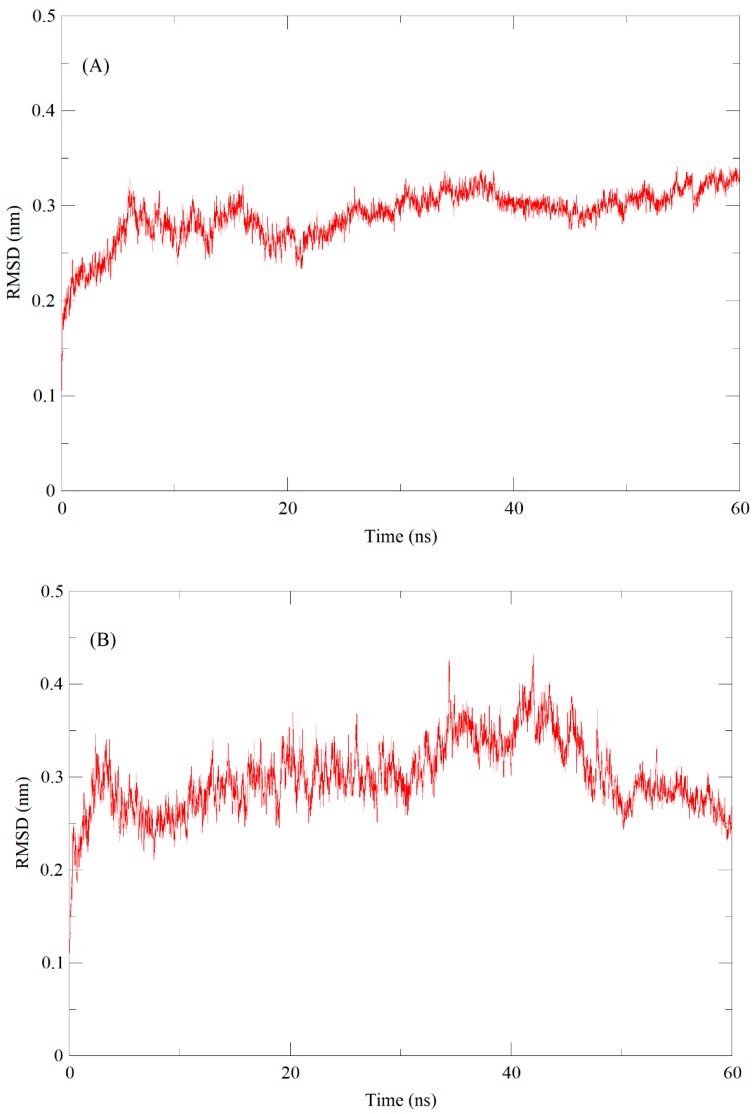
Molecular dynamics-based Epithiospecifier Protein (ESP) and Nitrile-specifier Protein (NSP) models refinement through 60 ns simulation of (**A**) ESP and (**B**) NSP backbone. RMSD is root mean square deviation.

**Figure 6 molecules-25-00772-f006:**
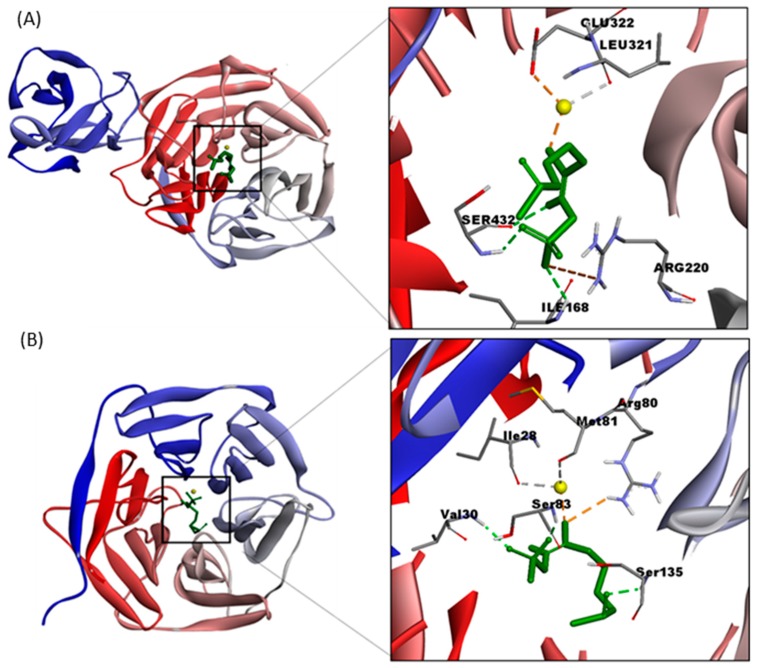
Three-dimensional structures of the complexes (**A**) NSP-4MSOB protonated at pH 5, and (**B**) ESP–4MSOB protonated at pH 3 and its interactions. In both complexes iron (Fe^2+^) was added as a cofactor (showed in yellow color).

**Table 1 molecules-25-00772-t001:** Identification of glucosinolates detected in broccoli inflorescences. t_R_ is the retention time and [M-H]^−^ is the m/z signal.

Peak	Glucosinolate	t_R_ (min)	[M-H]^−^	MS^2^ (m/z)	References
GI	Glucoiberin	3.8	422.3	357.8; 258.6; 274.6; 226.3; 197.1	[34,35,36]
SIN	Sinigrin	4.7	358.4	258.6; 161.3; 194.3; 274.4; 242.6	[35,36,37]
GRA	Glucoraphanin	6.1	436.4	371.9; 420.7; 258.7; 177.3; 290.7; 193.5	[34,35,36,37,38]
HGBS	Hydroxy-glucobrassicin	12.4	463.3	364.7; 266.7; 284.7; 159.3; 239.7; 259.3	[36,38]
GBS	Glucobrassicin	15.8	447.5	258.7; 274.7; 241.0; 205.0; 290.7; 260.7	[34,36,39]

**Table 2 molecules-25-00772-t002:** Binding affinity energy (kcal/mol) and affinity constant (k_A_, M) obtained from the Autodock Vina program, for aglycone from five glucosinolates (GLS) (4OHI3M, I3M, 4MSOB, 3MSOP, and 2PROP), interacting with the nitrile-specifier protein (NSP) or epithiospecifier protein (ESP) at different pH value (1, 3, 5, 7), using iron (Fe^2+^) as cofactor.

			Nitrile-Specifier Protein		Epithiospecifier Protein	
Precursor	Aglycone	pH	Binding Affinity Energy (kcal/mol)	k_A_ (M)	Binding Affinity Energy (kcal/mol)	k_A_ (M)
	4OHI3M	1	−8.7	2400	−7.0	140
Hydroxy-glucobrassicin		3	−8.6	2000	−5.7	15
		5	−8.5	1700	−6.8	96
		7	−8.9	3300	−5.9	21
	I3M	1	−8.5	1700	−7.3	220
		3	−8.6	2000	−7.3	220
Glucobrassicin		5	−8.6	2000	−7.3	220
		7	−8.5	1700	−7.4	270
	4MSOB	1	−6.5	58	−5.7	15
Glucoraphanin		3	−6.4	49	−5.9	21
		5	−6.7	82	−5.7	15
		7	−6.6	81	−5.6	13
	3MSOP	1	−6.2	35	−5.2	6.5
		3	−6.2	35	−5.4	9.1
Glucoiberin		5	−6.3	41	−5.5	11
		7	−6.3	41	−5.3	7.7
	2PROP	1	−6.0	25	−5.1	5.5
Sinigrin		3	−5.8	18	−5.2	6.5
		5	−5.8	18	−5.2	6.5
		7	−5.8	18	−5.0	4.6

**Table 3 molecules-25-00772-t003:** Interactions of the complexes NSP-4MSOB at pH 5, and ESP-4MSOB at pH 3. Structural details of NSP-4MSOB and ESP-4MSOB complexes are given in Appendix A.

Complex	Number	Interaction	Distance (Å)	H-Donor	H-Acceptor
NSP-4MSOB pH5	1	ARG220	5.36	A:ARG220:NH1	:4MSOB:O13
	2	GLU322	2.47	:4MSOB:Fe5	A:GLU322:OE2
	3	ILE168	2.75	A:ILE168:H	:4MSOB:O13
	4	SER432	2.06	A:SER432:H	:4MSOB:O12
	5	SER432	2.40	:4MSOB:H8	A:SER432:O
	6	LEU321	2.57	:4MSOB:Fe5	A:LEU321:O
Non-bonded interacting residues	SER165, HIS166, ALA169, GLN170, GLY218, VAL219, MET221, VAL222, GLY320, TRP329, HIS343, SER430, ALA431, THR433, SER434				
ESP-4MSOB pH3	1	VAL30	2.11	A:VAL30:H	:4MSOB:O12
	2	ARG80	2.93	A:ARG80:HH21	:4MSOB:S14
	3	SER83	1.98	A:SER83:H	:4MSOB:O11
	4	SER135	2.12	A:SER135:H	:4MSOB:O2
	5	ILE28	2.54	:4MSOB:Fe3	A:ILE28:O
	6	MET81	2.50	:4MSOB:Fe3	A:MET81:O
Non-bonded interacting residues:	HIS26, GLY27, VAL31, GLY32, GLY78, THR79, GLY85, MET133, ALA134, ASP136, GLU137, HIS322, GLU334, ASP333.				

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
