# Peer review of "Molecular Modeling of Epithiospecifier and Nitrile-Specifier Proteins of Broccoli and Their Interaction with Aglycones"

_molecules, 2020, doi:10.3390/molecules25040772_

Round 1
Reviewer 1 Report
In the manuscript ID: molecules-689960 the Authors describe the molecular modeling of ESP and NSP proteins of broccoli and their interactions with glucosinolates using molecular docking method. The level of the originality and general interest of this study is modest. However, I can recommend this manuscript for publication in Molecules after major improvement.
Comments and questions to the Authors:
Identification of molecular structures of glucosinolates detected in broccoli inflorescences should be supported by other spectroscopic mthods (e.g. NMR) or X-ray analysis. Are there significant differences in the structures (e.g. aminoacid sequences, 3D conformation) of modeled ESP and NSP proteins for broccoli inflorescences and X-ray structures of ESP (PDB: 5GQ0) and NSP1 (PDB: 5GQT) obtained from A. thaliana? Which means the statement (line 176): "These results agree with the resolved structures of ESP and NSP from A. thaliana". Whether it was necessary to model ESP and NSP proteins for docking of glucosinolates obtained from broccoli inflorecences and whether it could not be used for this purpose X-ray structures of ESP and NSP1 from A. thaliana? The molecular structures of 4OHI3M, I3M, 4MSOB, 3MSOP, and 2PROP should be presented in the manuscript. Moreover, numbering of atoms for 4MSOB should be shown (Table 3).
Author Response
Comments and Suggestions for Authors
In the manuscript ID: molecules-689960 the Authors describe the molecular modeling of ESP and NSP proteins of broccoli and their interactions with glucosinolates using molecular docking method. The level of the originality and general interest of this study is modest. However, I can recommend this manuscript for publication in Molecules after major improvement.
Comments and questions to the Authors:
Identification of molecular structures of glucosinolates detected in broccoli inflorescences should be supported by other spectroscopic mthods (e.g. NMR) or X-ray analysis.
R: NMR and X-ray are reliable methods for compounds identification and structure analyses, however HPLC-MS analysis is currently one of the most reliable methods and widely used in similar molecular studies. LC-ESI-MS/MS has been used in several studies focused on glucosinolates and similar compounds. In our work we used the method reported by Francisco et al., (J Chromatogr A, 1216(38), 6611-6619). Other recent works that used only LC-ESI-MS/MS methods for identification are the following:
Bertóti, R., et al., (2019). Variability of Bioactive Glucosinolates, Isothiocyanates and Enzyme Patterns in Horseradish Hairy Root Cultures Initiated from Different Organs. Molecules. 24(15): p. 2828.-Blažević, I., et al., (2019). Bunias erucago L.: Glucosinolate Profile and In Vitro Biological Potential. Molecules. 24(4): p. 741.
Are there significant differences in the structures (e.g. aminoacid sequences, 3D conformation) of modeled ESP and NSP proteins for broccoli inflorescences and X-ray structures of ESP (PDB: 5GQ0) and NSP1 (PDB: 5GQT) obtained from A. thaliana?
R: The consensus sequence obtained for broccoli NPS has 467 residues while that of A. Thaliana (Q9SDM9) has 470. The identity between both sequences was 80%. The model for broccoli ESP was built based on the reported amino acid sequence (Q4TU02). Simple alignment with Q8RY71 (A. thaliana) gave 76% identity. Accordingly the sequences differ from those from A. Thaliana and using the structures 5GQT and 5GQ0 in the docking simulations would most likely give different results. 80% or lower identity usually affects folding, orientation and conformation of the active or binding sites. This was explained in the manuscript.
Which means the statement (line 176): "These results agree with the resolved structures of ESP and NSP from A. thaliana". Whether it was necessary to model ESP and NSP proteins for docking of glucosinolates obtained from broccoli inflorecences and whether it could not be used for this purpose X-ray structures of ESP and NSP1 from A. thaliana?
R: This sentence was replaced by “Despite using structural models that are less reliable than crystallographic structures, conservation of the residues that belong to the substrate binding sites could be observed when comparing with ESP and NSP from A. thaliana.”
The molecular structures of 4OHI3M, I3M, 4MSOB, 3MSOP, and 2PROP should be presented in the manuscript. Moreover, numbering of atoms for 4MSOB should be shown (Table 3).
R: The molecular structures of 4OHI3M, I3M, 4MSOB, 3MSOP, and 2PROP were included in the manuscript (Fig. 2(C)). Numbering of atoms for 4MSOB and detailed structural information was included as supplementary material (Fig. S.6) because of space limitation; Fig. S.6 is referred to in Table 3.
Reviewer 2 Report
The paper "Molecular modeling of epithiospecifier and nitrile-specifier proteins of broccoli and their interaction with aglycones" by Román et al. deals with the interaction between a set of aglycones and NSP and ESP in broccoli.
The research design and the methods used are appropriate. The conclusions are well supported by the results.
Few misspellings (i.e. line 70 pag.3 "froom"). Please check all.
Author Response
Comments and Suggestions for Authors
The paper "Molecular modeling of epithiospecifier and nitrile-specifier proteins of broccoli and their interaction with aglycones" by Román et al. deals with the interaction between a set of aglycones and NSP and ESP in broccoli.
The research design and the methods used are appropriate. The conclusions are well supported by the results.
Few misspellings (i.e. line 70 pag.3 "froom"). Please check all
R: This was corrected in the text.
Reviewer 3 Report
In the manuscript titled “Molecular modeling of epithiospecifier and nitrile specifier proteins of broccoli and their interaction with aglycones” authors aimed to investigate the interaction of broccoli ESP and NSP with the aglycones derived from broccoli glucosinolates using molecular simulations. However the insilico experiments carried out for this study is inadequate and the presentation of the manuscript is not appropriate. So I would recommend a major revision for its current form.
Points to consider
For this study, authors have mentioned that they have taken MD based structure. But the modeled structure is just derived from 20ns simulation. For a protein of ~400 amino acids, how relevant can be a model with such lower a molecular simulation be?. The major conformational changes cannot be seen on this time scale. The authors need to perform a MD to a reasonable time scale to understand the structural changes in the binding region and in the whole protein. If not the authors can perform the complex simulation to further validate their results.
Few more MD based observations can also be reported like Hydrogen bond interaction and clustering analysis to identify major centroids of the reasonable clusters and a comparison would make the understanding better since the model is built based on prediction. It would be interesting if the pH based simulation is performed to support their Docking results to understand the residue level stability.
Why the authors have not used the newer force field of AMBER. Instead of old force field AMBER94. – Also the author says OPLS-AA/L all atom force filed used in page 11 L #291 what is the version – no reference given. The author needs to cite appropriately for all the computational tools that has been used in this study.
Figures need to be improved, for example the interaction part in figure 6 is not clear and need to be improved for better visualization.
Author Response
In the manuscript titled “Molecular modeling of epithiospecifier and nitrile specifier proteins of broccoli and their interaction with aglycones” authors aimed to investigate the interaction of broccoli ESP and NSP with the aglycones derived from broccoli glucosinolates using molecular simulations. However the insilico experiments carried out for this study is inadequate and the presentation of the manuscript is not appropriate. So I would recommend a major revision for its current form.
Points to consider
For this study, authors have mentioned that they have taken MD based structure. But the modeled structure is just derived from 20ns simulation. For a protein of ~400 amino acids, how relevant can be a model with such lower a molecular simulation be?. The major conformational changes cannot be seen on this time scale. The authors need to perform a MD to a reasonable time scale to understand the structural changes in the binding region and in the whole protein. If not the authors can perform the complex simulation to further validate their results.
R: We agree with longer molecular dynamics simulations can give more information, however we based our work on methodologies reported by several authors that coincide with the fact that 20 ns MD are appropriate to refine the models. Before conducting MD simulations, our models were already verified and validated by Verify3D and ERRAT. Time was chosen based on published articles that used GROMACS and proteins of similar length to ESP and NSP, such as Natarajan et al., (2015) who used 20 ns MD simulation for a 546 residues protein; Peymanfar et al., 2020 used 5-ns simulations for 200 residue-proteins; Tian, Gao, Chen, Liu, & Ju, 2019 performed 20 ns MD simulations in a 365-residues receptor.
Natarajan, S., Thamilarasan, S. K., Park, J.-I., Chung, M.-Y., & Nou, I.-S. (2015). Molecular Modeling of Myrosinase from Brassica oleracea: A Structural Investigation of Sinigrin Interaction. Genes, 6(4), 1315-1329.
Peymanfar Sh, Roghanian R, Ghaedi K, Zarkesh-Esfahani SH, Yari R. (2020). Characterization and in silico analysis of the structural features of G-CSF derived from lysates of Escherichia coli. Cell J. 21(4): 426-432.
Tian, Y., Gao, Y., Chen, Y., Liu, G., & Ju, X. (2019). Identification of The Fipronil Resistance Associated Mutations in Nilaparvata lugens GABA Receptors by Molecular Modeling. Molecules. 24(22), 4116.
Few more MD based observations can also be reported like Hydrogen bond interaction and clustering analysis to identify major centroids of the reasonable clusters and a comparison would make the understanding better since the model is built based on prediction.
R: We included the hydrogen bonds interactions (Fig. S.4) and the cluster analysis (Fig S.5). Discussion was complemented with this information.
It would be interesting if the pH based simulation is performed to support their Docking results to understand the residue level stability.
R: We thank the referee for this suggestion, currently we are working in the experimental validation of the results presented in this manuscript, and will perform the MD simulations at different pHs in order to help interpretation of the results and propose a mechanism. This will constitute another article with a more experimental approach.
Why the authors have not used the newer force field of AMBER. Instead of old force field AMBER94. – Also the author says OPLS-AA/L all atom force filed used in page 11 L #291 what is the version – no reference given. The author needs to cite appropriately for all the computational tools that has been used in this study.
R: We apologize for the confusion, it was a typing error in the abstract. In the Methods section it was described properly. This was corrected. The adequate references were included.
Figures need to be improved, for example the interaction part in figure 6 is not clear and need to be improved for better visualization.
R: All the figures were revised, Figs. 2, 3 and 6 were corrected.
Round 2
Reviewer 1 Report
I have no comments.
Author Response
We thank reviewer 2 for the comments and suggestions.
Reviewer 3 Report
The author's responses were acceptable with regard to all, except simulation time. I also request authors not to argue saying that this work has done only 5ns and the other work has done 20ns since we also justify that this is enough. Since computational technology is developing year by year and it is almost 5 years from the paper they quote.
Author Response
MD simulations of 60 ns were performed. Figure 5 amd Figs S.4 and S.5 were corrected accordingly, as well as the text.
Additionally, language was revised and improved.